# Modeling Synucleinopathy Using hESC-Derived Cerebral Organoids

**DOI:** 10.3390/cells14181436

**Published:** 2025-09-15

**Authors:** So Jin Kim, Won Hee Jung, Mu Seog Choe, Ye Seong Jeon, Min Young Lee

**Affiliations:** 1Department of Molecular Physiology, College of Pharmacy, Research Institute of Pharmaceutical Sciences, Vessel-Organ Interaction Research Center (VOICE, MRC), Kyungpook National University, Daegu 41566, Republic of Korea; ksjin0110@naver.com (S.J.K.); gardenhee99@gmail.com (W.H.J.); sung325000@gmail.com (Y.S.J.); 2Department of Genetics, Yale Stem Cell Center, Yale Child Study Center, Yale School of Medicine, New Haven, CT 06519, USA; museog.choe@yale.edu

**Keywords:** cerebral organoid, human pluripotent stem cells, synucleinopathy, disease modeling, drug testing

## Abstract

Animal and cellular models harboring *SNCA* gene mutations have been instrumental in synucleinopathy, but faithful human brain models remain limited. Here, we report the development of a human cerebral organoid (CO) model of synucleinopathy carrying the Ala53Thr mutation in *SNCA* (*SNCA^A53T^*). Using a human embryonic stem cell (hESC) line overexpressing *SNCA^A53T^* (A53T hESC line), we generated COs (A53T COs) that recapitulate hallmark features of synucleinopathy. These A53T COs exhibited elevated α-synuclein (α-Syn) expression, the increased phosphorylation of α-Syn, and Lewy body-like aggregations. Notably, we also observed the increased expression of phosphorylated tau and neurofibrillary tangle-like silver deposits, although amyloid β expression and accumulation remained unchanged. To evaluate the utility of this model in drug screening, we treated A53T COs with synuclean D (SynD), an inhibitor of α-Syn aggregation, which significantly reduced both α-Syn and tau phosphorylation without affecting total α-Syn levels. Together, our findings establish a robust hESC-derived synucleinopathy CO model harboring the *SNCA^A53T^* mutation, demonstrating its potential as a valuable tool for therapeutic drug screening.

## 1. Introduction

Synucleinopathies are a group of neurodegenerative diseases characterized by the abnormal accumulation of α-synuclein (α-Syn) aggregates in neurons or nerve fibers [1]. α-Syn is a presynaptic neuronal protein encoded by the *SNCA* gene, and plays roles in neurotransmitter release, synaptic transmission, vesicular transport, and fatty acid binding [2,3]. A growing body of in vitro and in vivo evidence suggests that the misfolding and aggregation of α-Syn is a central pathogenic event. Representative synucleinopathies include Parkinson’s disease (PD) and dementia with Lewy bodies (DLB) [4]. In these disorders, intracellular α-Syn aggregates are typically found in the form of Lewy bodies (LBs) or Lewy neurites (LNs) [5]. Multiple mechanisms contribute to α-Syn aggregation, including genetic mutations, specific cellular microenvironmental conditions, posttranslational modifications, and interactions with cellular membranes and other proteins [3,6,7,8]. Missense mutations in the *SNCA* gene have been shown to enhance α-Syn aggregation and disease severity [9].

Currently, most therapeutic strategies for synucleinopathies are aimed at designing drugs that directly target α-Syn and its aggregation pathway [10]. The development of such drugs and a deeper understanding of these diseases require effective disease models. Most studies modeling synucleinopathies utilize *SNCA* gene mutations. Known pathogenic missense mutations in *SNCA*, such as A53T, A30P, E46K, H50Q, and G51D, have been used successfully in both in vitro and in vivo studies [11,12]. In particular, the A53T mutation—the first identified *SNCA* missense mutation—has been shown to increase α-Syn aggregation and toxicity [13,14].

Both two-dimensional (2D) cell culture models and animal models have contributed to our understanding of synucleinopathies and have been valuable tools for drug development [15,16]. However, 2D models do not accurately recapitulate the in vivo microenvironment, histological structure, and cellular composition, and animal models present limitations related to genetic and physiological differences between humans and animals [17,18]. Recent advances in human pluripotent stem cell (hPSC)-derived cerebral organoid (CO) technology have provided a new avenue for studying human neurodegenerative diseases. These 3D structures self-assemble and partially recapitulate the cellular diversity, function, and architecture of the brain [19]. Currently, organoid systems are considered to occupy an intermediate position between 2D cell cultures and in vivo animal models, and are suggested as alternatives that may partially overcome the limitations of traditional models, such as the lack of natural tissue environment in 2D cultures and interspecies differences in animal models [19,20,21]. Human brain organoids have already been used to model a variety of neurological disorders with diverse approaches to disease modeling and downstream applications [22]. Recently, various synucleinopathy models targeting PD have been developed using the brain organoid technology with the combination of specific genetic mutations. Human midbrain organoid models carrying a triplication of the *SNCA* gene were reported and they showed PD-related pathologies such as elevated α-Syn levels, α-Syn aggregation, LB-like inclusions as well as loss of dopaminergic neurons [23,24,25]. Additionally, other types of genetic mutations such as G2019S LRRK2 mutation, PINK1 knockout, L444P GBA1 mutation have been used for PD organoid modeling. They also recapitulated PD pathologies including increased α-Syn aggregation, mitochondrial dysfunction, etc. [26,27,28,29]. In this study, we established a CO synucleinopathy model using the A53T α-Syn mutation. The A53T mutation was first reported in a human patient with familial PD, and its pathological association in humans has been confirmed [30,31]. It has been validated in various cell and animal models, so when used in a CO model, it is expected to provide a reliable synucleinopathy model [32,33,34]. We confirmed the expression of several types of pathology in this model and also demonstrated its potential use for drug testing. We expect that this model will serve as a valuable tool for studying diseases accompanied by synucleinopathy, such as PD or DLB.

## 2. Materials and Methods

Human embryonic stem cell (hESC) culture. The human embryonic stem cell (hESC) line SNUhES31 was obtained from the Institute of Reproductive Medicine and Population, Medical Research Center, Seoul National University Hospital, Republic of Korea. hESCs were cultured on 10 μg/mL mitomycin-C (Roche, Mannheim, Germany)-treated mouse embryonic fibroblasts in hESC medium containing 20% knockout serum replacement (Life Technologies, Carlsbad, CA, USA), 1% minimum essential media-nonessential amino acids (MEM-NEAA) (Life Technologies, Carlsbad, CA, USA), 1% GlutaMAX (Life Technologies, Carlsbad, CA, USA), and 7 μL/L β-mercaptoethanol (Sigma-Aldrich, St. Louis, MO, USA), all in DMEM/F-12 with 20 ng/mL bFGF (R&D Systems, Minneapolis, MN, USA). For feeder-free culture, hESCs were detached from the feeder cells using 1 mg/mL diapase (Life Technologies, Carlsbad, CA, USA) and maintained in mTeSRTM1 medium (STEM CELL Technologies, Vancouver, BC, Canada) on Geltrex (Life Technologies, Carlsbad, CA, USA)-coated culture plates. Cells were subcultured as small clusters every 4 days using a 0.5 mM EDTA solution.

The establishment of the hESC line with the Ala53Thr *SNCA* mutation (*SNCA^A53T^*): To generate the *SNCA^A53T^* mutant hESC line, the gene harboring the Ala53Thr mutation (GCA → ACA) was synthesized (Macrogen Inc., Seoul, Republic of Korea) and cloned into a lentiviral plasmid under the CAG promoter (*GAG-SNCA^A53T^*). The vector was sequence-verified (Enzynomics, Daejoen, Republic of Korea). For virus production, the plasmid, along with envelope (VSVg) and packaging (gag-pol) plasmids, was co-transfected into 85% confluent HEK293T cells (ATCC) in a 10 cm dish using the X-tremeGENE^TM^ HP transfection reagent (Roche). Medium was replaced after 24 h, and 10 mL of virus-containing supernatants were collected daily for three days, pooled (30 mL), and concentrated to 200 µL by ultracentrifugation (25,000 rpm, 4 °C, 2 h; Hitachi, Tokyo, Japan). Feeder-free hESCs in a 12-well plate were transduced with the virus (2 × 10^7^ IU/mL) in Essential-8 medium plus 2 µg/mL hexadimethrine bromide (Sigma-Aldrich, St. Louis, MO, USA) for three days, with daily medium changes. Afterward, 500 transfected hESCs were plated onto an STO cell-coated 6-well plate (Corning, Corning, NY, USA). After 7 days of culture, single-cell–derived colonies were manually dissected and transferred to Matrigel-coated 24-well plates. Colonies were expanded under feeder-free conditions. Expression of α-Syn was confirmed by Western blotting, and the transfected cells were designated as the A53T hESC line.

Generations of cerebral organoids: Cerebral organoids were generated from hESCs using a previously described protocol [35]. Briefly, feeder-free hESCs were dissociated into single cells by incubation with 0.5 mM EDTA for 4 min and Accutase^®^ for 4 min, then seeded at 9 × 10^3^ cells per well in ultra-low-attachment 96-well U-bottom plates with low-bFGF hESC medium [1% MEM-NEAA (Thermo Fisher Scientific, Waltham, MA, USA), 1% GlutaMAX, 7 μL/L β-mercaptoethanol, 20% knockout serum replacement in DMEM/F-12], 5 ng/mL bFGF, and 50 μM Y27632, a Rho-associated protein kinase inhibitor. This was designated day 0. The media were changed every other day. On day 2, the medium was replaced with low-bFGF hESC medium containing 50 μM Y27632. On day 4, medium without Y27632 was used. On day 6, embryoid bodies (EBs) were transferred to ultra-low-attachment 24-well plates with neural induction medium [1% MEM-NEAA, 1% GlutaMAX, 1% N_2_ supplement (Thermo Fisher Scientific), 1 μg/mL heparin (Sigma-Aldrich) in DMEM/F-12]. On day 8, fresh neural induction medium was added. On day 10, neuroepithelial tissues were transferred to Matrigel droplets in CO differentiation medium without vitamin A [0.5% MEM-NEAA, 1% GlutaMAX, 1% B27 supplement (minus vitamin A), 0.5% N_2_ supplement, 2.5 μg/mL human insulin (Roche), 3.5 μL/L β-mercaptoethanol in a 1:1 mixture of neurobasal medium (Thermo Fisher Scientific)]. On day 12, droplets were placed in fresh CO differentiation medium without vitamin A. On day 14, organoid-containing Matrigel droplets were transferred to a spinner flask (Corning) with cerebral organoid differentiation medium [0.5% MEM-NEAA, 1% Glutamax, 1% B27 supplement, 0.5% N2 supplement, 2.5 μg/mL human insulin (Roche), and 3.5 μL/L β-mercaptoethanol in a 1:1 mixture of neurobasal medium (Thermo Fisher Scientific)] and subsequently maintained with medium changes every 7 days.

Immunocytochemistry: Dissociated cells were fixed with 4% paraformaldehyde (PFA) solution in PBS (Wako, Osaka, Japan), then permeabilized and blocked with 10% normal goat serum (NGS) (Vector Laboratories, Newark, CA, USA) in 0.1% PBST (vol/vol, Triton X-100 in PBS). Cells were incubated with primary antibodies in PBST containing 2% NGS for 24 h at 4 °C. Primary antibodies against the following proteins were used: NANOG (rabbit, Cell Signaling Technology, Danvers, MA, USA, 1:200), OCT4 (mouse, Santa Cruz, Dallas, TX, USA, 1:100), Tra-1-60 (mouse, Santa Cruz, 1:100), and SSEA4 (mouse, Santa Cruz, 1:100). After three washes with PBST, the cells were incubated with secondary antibodies (goat anti-rabbit or anti-mouse Alexa Fluor 488/647; Thermo Fisher Scientific, 1:500) for 3 h at room temperature. Images were acquired with a Leica TCS SP5 II confocal microscope.

Immunohistochemistry: COs were fixed with a 4% PFA solution in PBS, cryoprotected in 15% and 30% sucrose, embedded in OCT (Leica, Wezlar, Germany), and cryosectioned into 10-μm-thick sections using a cryotome (CM1850 cryostat; Leica). For immunohistochemistry, the sections were permeabilized and blocked with 10% NGS in PBST, then incubated with primary antibodies in PBST containing 2% NGS for 24 h at 4 °C. Primary antibodies against the following proteins were used: SOX2 (rabbit, Cell Signaling Technology, 1:200), TUJ 1 (rabbit, Cell Signaling Technology, 1:200), DCX (rabbit, Cell Signaling Technology, 1:200), MAP2 (rabbit, BioLegend, San Diego, CA, USA, 1:200), VGluT1 (rabbit, GeneTex, Irvine, CA, USA, 1:200), GFAP (rabbit, Cell Signaling Technology, 1:200), α-synuclein (mouse, Santa Cruz, 1:100), Phospho-Tau (Ser202, Thr205) (AT8) (mouse, Thermo Fisher Scientific, 1:200), 4G8 (mouse, BioLegend, 1:200), 6E10 (mouse, BioLegend, 1:200), VGluT2 (rabbit, Cell Signaling Technology, 1:200), DLX1 (rabbit, Cell Signaling Technology, 1:200), and GAD1/2 (mouse, Santa Cruz Biotechnology, 1:100). After three washes with PBST, sections were incubated with secondary antibodies (goat anti-rabbit or anti-mouse Alexa Fluor 488/647, Thermo Fisher Scientific, 1:500) for 3 h at room temperature. Images were obtained by confocal microscopy (Leica TCS SP5 II; Leica).

For diaminobenzidine (DAB) staining, sections blocked with 10% NGS in PBST were incubated with the primary antibody, anti-phospho-α-synuclein (Ser129) (mouse, Santa Cruz Biotechnology, 1:100) for 24 h at 4 °C, then with biotinylated horse anti-mouse secondary antibody (Vector Laboratories) for 3 h at room temperature. Antibody detection was performed using the avidin-biotin peroxidase system (Vector Lab) and DAB substrate (Vector). Sections were counterstained with hematoxylin (4 min) and imaged with a Pannoramic SCAN II (3D Histech, Budapest, Hungary) light microscope (TM-20B, Taeshin Bio Science, Nam-yangju, Republic of Korea).

Western blot analysis: COs were sonicated in RIPA buffer (iNtRON Biotechnology, Seongnam, Republic of Korea) on ice (Vibra-Cell^TM^, Newtown, CT, USA). Extracted proteins were separated by 10–12% SDS-PAGE and transferred to PVDF membranes (Millipore, Burlington, MA, USA). Membranes were washed with TBST (150 mM NaCl, 10 mM Tris-HCl pH 7.6, 0.1% Tween-20), blocked with 5% skim milk (Millipore) for 1 h, and incubated with primary antibodies for α-synuclein (1:1000) and phospho-Tau (Ser202, Thr205) (AT8) (1:1000). After TBST washes, horseradish peroxidase-linked goat anti-mouse IgG secondary antibody (Santa Cruz, 1:5000) was applied. Bands were visualized using enhanced chemiluminescence (Thermo Fisher) and imaged with a Bio-Rad Chemidoc^TM^ imaging system (Bio-Rad, Hercules, CA, USA). Densitometry was performed using ImageJ software (version 1.41) (National Institutes of Health, Bethesda, MD, USA).

Enzyme-Linked ImmunoSorbent Assay (ELISA): Phospho-α-Syn levels were quantified using a P-SNCA ELISA kit (MyBioSource, San Diego, CA, USA). COs were lysed by sonication in RIPA buffer on ice; protein concentrations were determined by BCA assay. ELISA was performed using 10 μg lysate according to the manufacturer’s protocol. Absorbance at 450 nm was measured with a microplate reader (Infinite M200 Pro, Tecan, Männedorf, Switzerland).

Bielschowsky’s silver staining: Silver staining was performed using the VitroView^TM^ Bielschowsky’s Silver Stain kit (VitroVivo Biotech, Rockville, MD, USA). Briefly, the cryosectioned slides were incubated in a pre-warmed (40 °C) silver nitrate solution for 15 min, rinsed with distilled water (DW), and then incubated in ammonium silver solution at 40 °C for 30 min. The slides were treated with Developer Stock Solution for 1 min, and subsequently the reaction was halted by immersing the slides in 1% ammonium hydroxide solution for 1 min. After three washes with DW, the slides were placed in 5% sodium thiosulfate solution for 5 min, then washed three times with DW, dehydrated in 100% ethanol, cleared with xylene, and mounted with coverslips using mounting medium. Images were obtained using Pannoramic SCAN II (3D Histech) and a conventional light microscope (TM-20B, Taeshin Bio Science).

Statistical analysis: All data are presented as mean ± standard deviation (S.D.). Differences between the mean values were analyzed using Student’s *t*-test. A *p*-value < 0.05 was considered statistically significant.

## 3. Results

### 3.1. Establishment of A53T hESC Line and Characterization of A53T COs

We designed and generated lentiviral constructs to overexpress the *SNCA* gene with A53T mutation (*SNCA^A53T^*) under control of the CMV early enhancer/chicken β-actin (CAG promoter) (Figure 1). After synthesizing and inserting *SNCA^A53T^* into a lentiviral plasmid vector containing CAG, we transfected hESCs with the generated A53T lentiviral particles. Six single-cell-derived A53T hESC colonies were manually selected and expanded, as shown in Figure 1A. Among the six cell lines, one fAD-S hESC line (#6) was selected based on the α-Syn expression level in comparison with wild type (WT) by Western blotting (Figure 1B). Immunofluorescence staining of hPSC markers such as OCT4, NANOG, SSEA-4, and TRA-1-60 confirmed that the transduction of the *SNCA^A53T^* gene did not affect the undifferentiated state of the A53T hESC line (Figure 1C,D).

Using this established synucleinopathy hESC line, we generated COs and evaluated their formation capacity at 150 days (Figure 1E). Expression analysis of regional brain markers revealed that A53T COs, like WT COs, exhibited typical markers of COs, including a neural progenitor markers such as (SOX2, Sex determining region Y-box 2), a neuronal marker (TUJ1, beta-Tubulin III), a neuronal marker (MAP2, Microtubule-associated protein 2), a neuronal marker (DCX, Doublecortin), astrocyte marker (GFAP, Glial fibrillary acidic protein), a cortical glutamatergic neuron marker (vGluT1, Vesicular glutamate transporter 1), a subcortical glutaminergic neuron marker (vGluT2, Vesicular glutamate transporter 2), an interneuron precursor marker (DLX1, Distal-less homeobox 1), and an inhibitory neuron identity marker (GAD1/2, Glutamate decarboxylase 1/2). (Figure 1F). These results confirmed that α-Syn^A53T^ gene transduction did not impair normal CO formation.

### 3.2. Expression of Phospho-αSyn-Related Pathologies in A53T COs

To verify pathological phenotypes of synucleinopathy, we assessed the expression of α-Syn and phospho-α-Syn in 150-day-old A53T COs. As shown in Figure 2A,B, western blot analysis and ELISA demonstrated significantly elevated levels of both α-Syn and phospho-α-Syn in A53T COs compared to WT COs. These increases were also consistently observed by immunostaining (Figure 2C). DAB staining revealed phospho-α-Syn-positiveLewy body-like inclusions in A53T COs (Figure 2D). Additionally, IHC staining showed SQSTM1/p62 apparently accumulated at phospho-α-Syn-positive inclusions. It is well known that p62/SQSTM1, an adaptor protein, is a component of Lewy bodies [36]. These results indicate successful recapitulation of phospho-α-Syn-related pathologies of synucleinopathy.

### 3.3. Confirmation of Phospho-Tau and Amyloid-β Expression Levels in LBD COs

Although synucleinopathy is primarily characterized by α-Syn aggregation, neurodegenerative diseases often exhibit additional pathological phenotypes, including the hyperphosphorylation of tau protein or the accumulation of Aβ [37]. We examined whether these pathological markers are expressed in a 150-day-old A53T CO model. As shown in Figure 3A, the expression levels of phospho-tau (AT8) were significantly higher in A53T COs than WT COs, a finding further confirmed by immunohistochemistry (Figure 3B). Furthermore, Bielschowsky’s Silver staining also revealed NFT-like deposits in A53T COs, which were absent in WT COs (Figure 3C). We also examined amyloid β pathology by measuring the expression levels of Aβ 1–40 and 1–42 isoforms in 150-day-old WT and A53T COs using ELISA. As shown in Figure 3D, we did not observe differences in the levels of Aβ 1–40 and 1–42 between WT and A53T COs. These results were also observed in IHC images (Figure 3E).

### 3.4. Effects of α-Syn Aggregation Inhibitor in A53T CO Model

To evaluate the potential of the A53T CO model as a tool for drug screening, we examined the effects of SynuClean D (SynD), an established α-Syn aggregation inhibitor. COs were treated with 100 μM SynD from day 70 to day 150 during generation. ELISA analysis and immunofluorescence images revealed that SynD significantly reduced phospho-α-Syn levels in A53T COs (Figure 4A,B), while total α-Syn expression remained unchanged (Figure 4C). Notably, SynD treatment also significantly decreased phospho-tau (AT8) expression levels (Figure 4D).

## 4. Discussion

Synucleinopathy encompasses a group of neurodegenerative diseases characterized by abnormal accumulation of α-synuclein protein in the brain. These disorders, collectively known as synucleinopathies, include Parkinson’s disease (PD), dementia with Lewy bodies (DLB), multiple system atrophy (MSA), and rare disorders such as various neuroaxonal dystrophies [1]. Various mutations in the *SNCA* gene have been identified as causative factors in synucleinopathies, leading to the aggregation of α-Syn protein in the brain. Among these mutations, A53T was first discovered in familial PD patients from Greece, with affected individuals exhibiting extensive pathologies related to the α-Syn aggregation throughout the brain [30,38]. Given that human A53T α-Syn demonstrates high aggregation potential [39,40], we selected the A53T *SNCA* mutation for our CO synucleinopathy model.

Phosphorylation at Ser-129 serves as a specific marker for all synucleinopathy lesions, occurring after initial protein aggregation [41]. These intracellular α-Syn aggregates typically manifest as Lewy bodies (LBs) or Lewy neurites (LNs) [5]. LBs are primarily composed of abnormal filamentous aggregates resulting from mutations in the α-Syn protein [42], with phosphorylated α-Syn comprising the majority of these aggregates [43]. Our A53T COs demonstrated significantly elevated α-Syn phosphorylation, along with distinct phospho-α-Syn aggregation and LB-like inclusions, indicating successful recapitulation of key pathological phenotypes of synucleinopathy.

Notably, our A53T COs exhibited increased phospho-tau levels and neurofibrillary tangles (NFTs)-like silver deposition, suggesting potential involvement in the induction of tauopathy. This observation aligns with previous research demonstrating connections between A53T synucleinopathy and tauopathy. Wills et al. demonstrated increased hyperphosphorylated tau (p-tau) in A53T α-Syn mice [44]. Teravskis et al. reported that A53T mutant α-Syn induces tau-dependent pathology expression [28]. Indeed, α-Syn pathology can trigger tau pathology, with both frequently coexisting in PD and DLB [45]. While these diseases often display co-aggregation of amyloidogenic proteins, including amyloid β [46], our A53T COs showed no increased Aβ expression. This suggests that the A53T mutation does not directly influence Aβ formation, despite the frequent co-existence of these pathologies in many neurodegenerative diseases [37,47]. We concluded that our A53T CO model recapitulates synucleinopathy and its related tau pathology.

To validate the utility of our A53T CO model for drug screening, we tested Synuclean D (SynD), a small molecule known to inhibit α-Syn aggregation and disassemble pre-formed aggregates [48,49]. Treatment with SynD significantly reduced both phospho-α-Syn and phospho-tau levels in A53T COs, demonstrating the potential effectiveness of this model for drug screening applications.

## 5. Conclusions

The exponential increase in neurodegenerative diseases, corresponding with increased human longevity worldwide, presents serious socioeconomic challenges. Despite this growing burden, effective treatments remain elusive, partly due to the lack of appropriate experimental models relevant to human pathology. While organoid technology cannot entirely replace traditional 2D cell cultures or animal models, it offers a valuable tool to bridge the gaps in existing models and provides additional insights into disease mechanisms [50]. Our synucleinopathy CO model using A53T *SNCA* gene mutation represents a promising platform for understanding the mechanisms underlying the pathogenesis of synucleinopathy and facilitating drug development.

## Figures and Tables

**Figure 1 cells-14-01436-f001:**
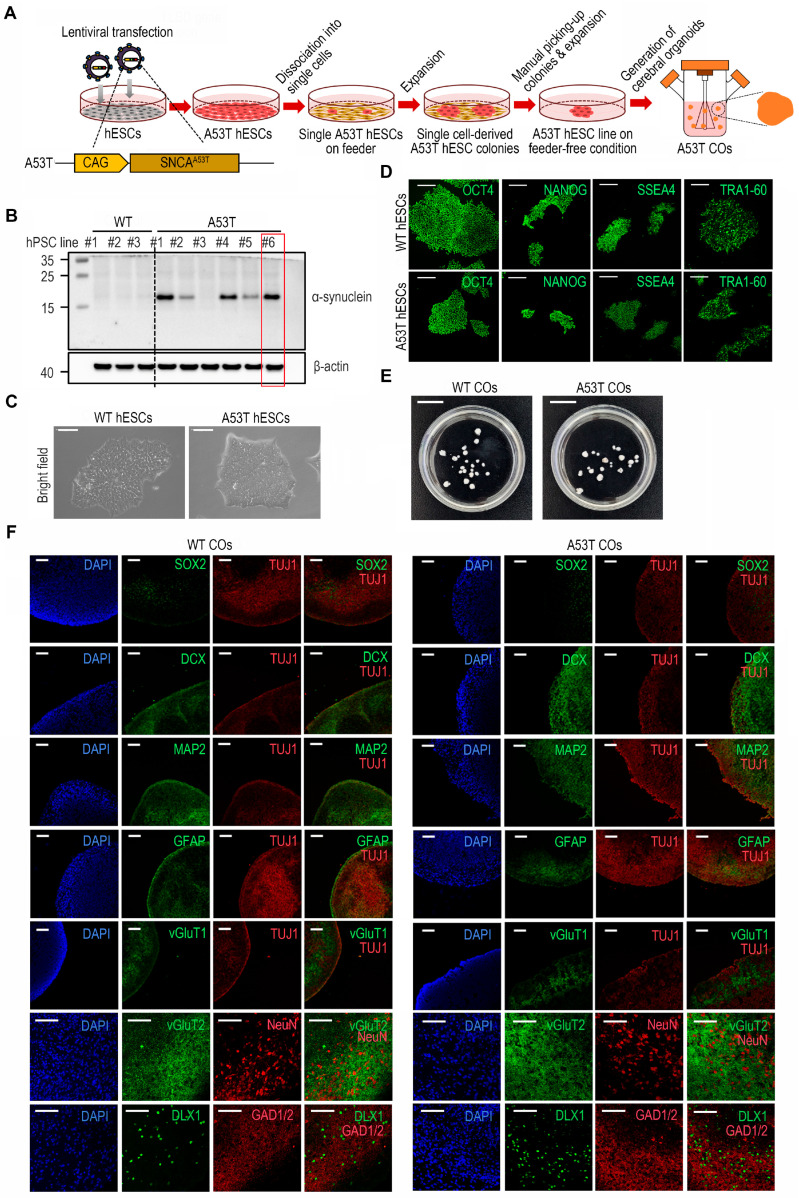
Establishment of a synucleinopathy human embryonic stem cell (hESC) line with the *SNCA^A53T^* mutation and generation of cerebral organoids (COs). (**A**) Procedural schematic for establishing synucleinopathy hESC lines and subsequent CO generation. The process includes: lentiviral transfection of hESCs with *SNCA^A53T^* mutation (A53T), single-cell plating on a feeder layer, colony selection, and CO generation. CAG: CMV early enhancer/chicken β-actin. (**B**) Western blot analysis comparing α-synuclein (α-Syn) expression levels between WT and A53T hESC lines. Selected A53T hESC lines indicated by red squares. (**C**) Brightfield microscopy of WT and A53T hESCs. Scale bar: 200 μm. (**D**) Immunofluorescence visualization of pluripotency markers (Oct4, Nanog, SSEA4, and Tra-1-60) in WT and A53T hESCs. Scale bar: 200 μm. (**E**) Morphological comparison of 150-day-old WT and A53T COs. Scale bar: 1 cm. (**F**) Neural identity markers in WT and A53T COs visualized by immunohistochemistry. Nuclear localization indicated by DAPI staining (blue). Scale bar: 100 μm.

**Figure 2 cells-14-01436-f002:**
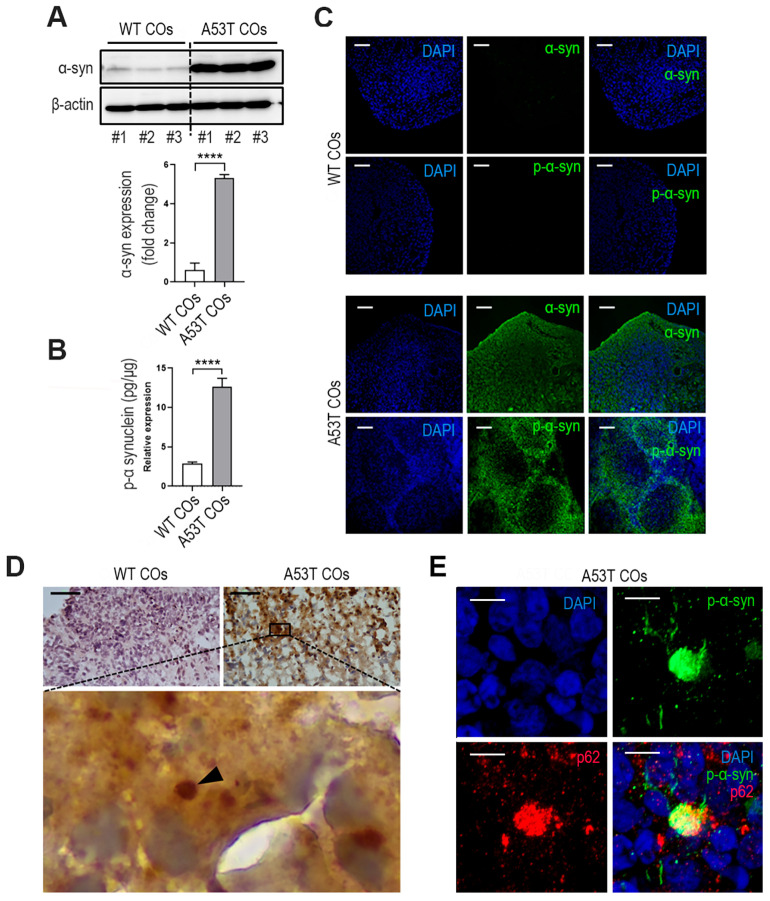
Synucleinopathy pathological markers in A53T CO model. (**A**) Western blot quantification of α-synuclein (α-Syn) in 150-day-old WT and A53T CO lysates. Mean ± SD; **** *p* < 0.0001 vs. WT; *n* = 3 per sample. (**B**) ELISA measurement of phospho-α-Syn levels in WT and A53T CO lysates at day 150. Mean ± SD; **** *p* < 0.0001 vs. WT; *n* = 3 per sample. (**C**) Representative immunofluorescence images of α-Syn and phospho-α-Syn in WT and A53T COs. Scale bar: 100 μm. (**D**) Lewy body-like inclusions in 150-day-old A53T COs visualized by DAB staining of phospho-α-Syn (Ser129) (black arrowhead). Scale bar: 50 μm. (**E**) High-magnification immunofluorescence image of phospho-α-Syn and SQSTM1/p62 in A53T CO at day 150. Scale bar: 5 μm.

**Figure 3 cells-14-01436-f003:**
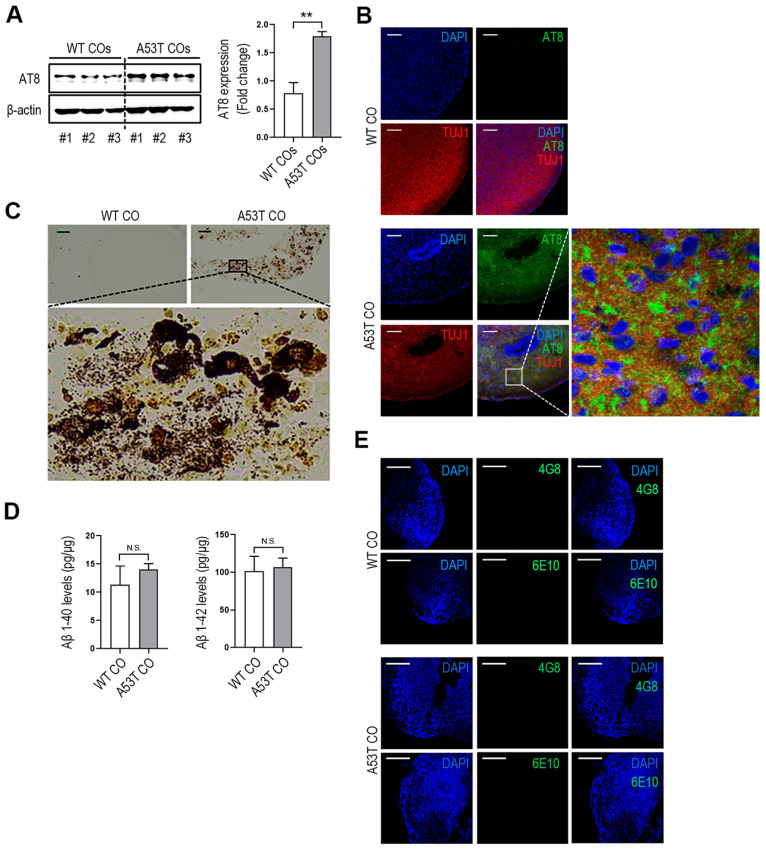
Expression of pathological markers of tauopathy in A53T CO model. (**A**) Western blot analysis of phospho-tau (AT8) expression in WT and A53T COs at day 150. Mean ± SD; ** *p* < 0.01 vs. WT; *n* = 3 per sample. (**B**) Representative immunofluorescence images of AT8 and Tuj1 in 150-day-old WT and A53T COs. Scale bar: 100 μm. (**C**) Bielschowsky’s silver staining revealing neurofibrillary tangles (NFTs)-like deposits in 150-day-old WT and A53T COs. Scale bar: 50 μm. (**D**) Quantification of Aβ1–40 and Aβ1–42 levels in the lysates of WT and A53T COs at day 150. Mean ± SD; N.S., not significant; *n* = 3 per sample. (**E**) Representative immunohistochemistry images of Aβ in 150-day-old WT and A53T COs. Scale bar: 200 μm.

**Figure 4 cells-14-01436-f004:**
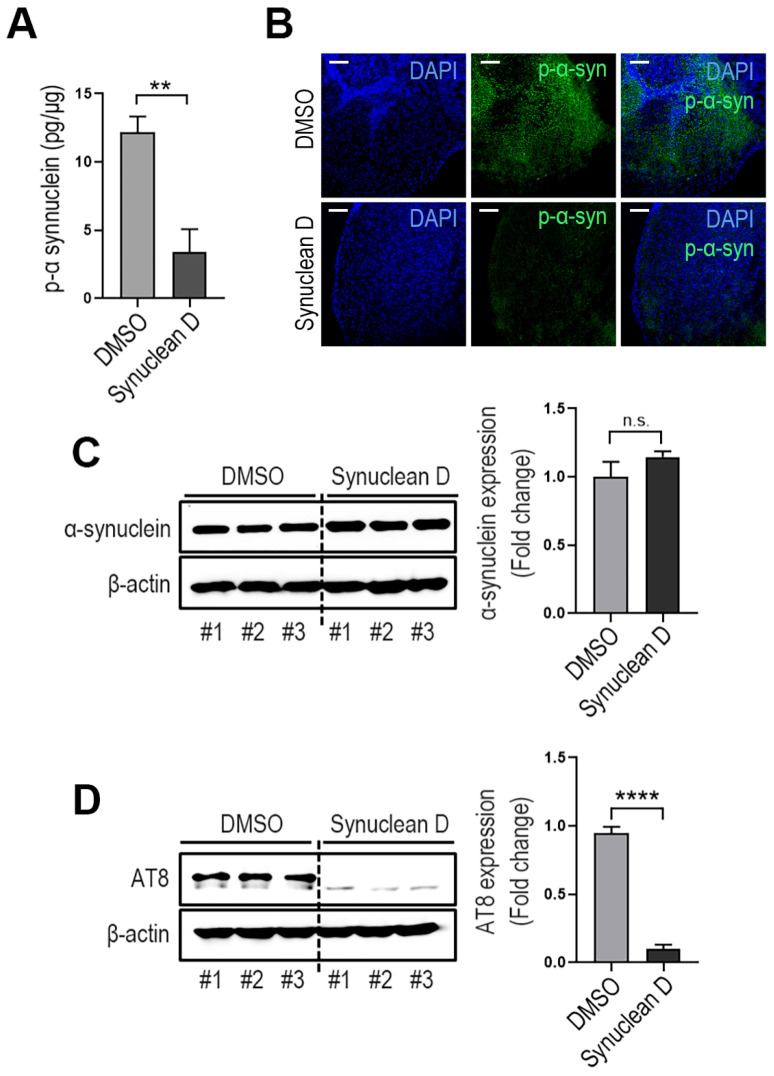
Evaluation of drug activity using the A53T CO model. (**A**) Effects of synucleanD on phospho-α-Syn levels in 150-day-old A53T COs. Mean ± SD; ** *p* < 0.01 vs. DMSO; *n* = 3 per sample. (**B**) Representative immunofluorescence images showing the phopho-α-Syn in A53T COs treated with or without Synuclean D. Scale bar: 100 μm. (**C**) Western blot analysis of synuclean D effects on total α-Syn levels in lysates of 150-day-old A53T COs. n.s.: not significant; *n* = 3 per sample. (**D**) Effect of synuclean D in phospho-tau levels (AT8) in 150-day-old A53T COs. Mean ± SD; **** *p* < 0.0001 vs. DMSO.

## Data Availability

The data presented in this study are available in the article.

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
