# Peer review of "Modeling Synucleinopathy Using hESC-Derived Cerebral Organoids"

_cells, 2025, doi:10.3390/cells14181436_

Round 1

Reviewer 1 Report

Comments and Suggestions for Authors

In this paper Kim et al. generated a human cerebral organoid (CO) model of synucleinopathy carrying the Ala53Thr mutation in SNCA and showed increased expression of alpha-synuclein expression, increased phosphorylation of alpha-Syn and tau, Lewy body-like aggregations and neurofibrillary tangle-like silver deposits in the CO carrying the Ala53Thr mutation.

  1. In the introduction, please summarize the previous reports on generation of organoids to model Parkinson disease.
  2. As in the introduction, this reviewer agrees that organoid systems are considered to occupy an intermediate position between 2D cell cultures and in vivo animal models, and are suggested as alternatives that may partially overcome the limitations of traditional models. In this sense please discuss advantage of the CO carrying the Ala53Thr mutation.
  3. In Fig. 2, appearance of the Lewy body-like inclusions should be further confirmed by several ways, for example double staining with SQSTM1/p62 or other inclusion markers or electron microscopy.
  4. In Fig. 3, please show higher-magnification immunofluorescent images of AT8-positive structures in the CO carrying the Ala53Thr mutation.
  5. In Fig. 4, please show western blotting and immunofluorescent images of phospo-alpha synuclein.

Reviewer 2 Report

Comments and Suggestions for Authors

The authors generated human cerebral organoids (with and without Ala53Thr mutation) and analyzed these structures. The authors use immunocytochemistry, immunohistochemistry, Bielschowsky’s silver staining and ELISA techniques and battery of Western blots. The topic is likely to attract the interest of many readers and fits well within the journal's scope. The manuscript contains a large amount of description characterising the validity of the 3D model system.

Abstract is well-written and nicely summarized the main findings of the manuscript. The introduction provides an objective summary of the background to the synucleinopathies and related neurodegenerative disorders [Parkinson’s disease (PD) and dementia with Lewy bodies (DLB)]. The authors highlighted the advantages and disadvantages of 2D models. Additionally, they explained why 3D cultures are much more relevant in this field. They objectively argue for the use of 3D models for investigation of neurodegenerative disorders. I really appreciate the introduction part of the manuscript. I was pleased to read that they explained in sufficient details how the COs were generated in Material and Methods section. The results section is easy to understand. Basically, well-designed experiments have been carried out to thoroughly test the 3D model system. The data presentation is consistent throughout the manuscript. The graphs/pictures really help to understand the written text. Discussion part is also well-written and the conclusion is relevant for the scientific community.

For improvement of the manuscript some points should be addressed:

1, Next, the authors should try if they can solve the tissue clearing and light-sheet microscopy for better imaging. As I know, it’s very expensive technique and needed lot of long trial for optimalization. It is worth for publication in top-tier journal.

2, Please, indicate the concentration/dilution of the (primary and secondary) antibodies in all cases.

3, What is the reason for the different alpha-synuclein expression in the six different Ala53Thr CO lines (Fig. 1B)?

4, In line 210. is a typo - correct: COs

5, Missing scale bar in Fig. 1E.

6, Why were organoids 150 days old chosen for the studies? Would we see similar changes in older/younger organoids?

7, I realized that in the Fig. 1F right panel the GFAP signal is significantly different compared to wild-type COs. The upper part of the tissue structure is GFAP negative. Why?

8, I am a bit puzzled related to Fig. 2E. Are you sure in green dot-like structure is enriched p-alpha-synuclein? Please also show the absence of this in wild-type COs. Why is it only found the p-alpha-synuclein signal in several neurons?

9, Why are shorter exposure times not used to develop membranes? I see, there are a lot of oversaturated pixels.

10, In line 316. is a typo - correct: drug development

11, The text (A53T CO – in the bottom; above the legend) of Fig. 3 is invisible.

Reviewer 3 Report

Comments and Suggestions for Authors

In this manuscript, the authors generated a human embryonic stem cell (hESC) line overexpressing the A53T mutant form of SNCA, and subsequently used it to produce cerebral organoids (COs). The A53T COs recapitulate hallmark features of synucleinopathy, including elevated α-synuclein (α-Syn) expression, increased phosphorylation of α-Syn at Ser129, and the presence of Lewy body-like aggregates. The study also demonstrates that SynuClean-D (SynD), an inhibitor of α-Syn aggregation, effectively reduces both α-Syn and tau phosphorylation, showing the application potential of this line for drug screening. Overall, this is a well-designed and well-written manuscript. 

One limitation, however, is that the A53T hESC line was generated through lentivirus-mediated overexpression of SNCA A53T, which represents an artificial system rather than a patient-derived model. Although the authors have shown that A53T overexpression does not affect hESC pluripotency at the iPSC stage or general neuronal differentiation in COs based on immunostaining, cerebral organoids contain a diverse population of neuronal subtypes. Thus, a potential concern is whether A53T overexpression might influence the development of specific neuronal subtypes.

To address this, I suggest performing bulk RNA sequencing to compare global gene expression profiles, including neuronal subtype marker expression, between control and A53T COs. This would strengthen the conclusion that overexpression of A53T does not bias neuronal specification within the organoids.

Round 2

Reviewer 1 Report

Comments and Suggestions for Authors

The Authors have addressed almost all of my comment on the original manuscript. The revised manuscript could be ready for publication.